# Evaluation of a Novel Thiol–Norbornene-Functionalized Gelatin Hydrogel for Bioprinting of Mesenchymal Stem Cells

**DOI:** 10.3390/ijms23147939

**Published:** 2022-07-19

**Authors:** Vadym Burchak, Fritz Koch, Leonard Siebler, Sonja Haase, Verena K. Horner, Xenia Kempter, G. Björn Stark, Ute Schepers, Alisa Grimm, Stefan Zimmermann, Peter Koltay, Sandra Strassburg, Günter Finkenzeller, Filip Simunovic, Florian Lampert

**Affiliations:** 1Department of Plastic and Hand Surgery, Faculty of Medicine, Medical Center, University of Freiburg, Freiburg, Hugstetter Strasse 55, 79106 Freiburg, Germany; vadym.burchak@uniklinik-freiburg.de (V.B.); leonard.siebler@uniklinik-freiburg.de (L.S.); verena.kassiopeia.horner@uniklinik-freiburg.de (V.K.H.); stark.bjoern@googlemail.com (G.B.S.); sandra.strassburg@uniklinik-freiburg.de (S.S.); guenter.finkenzeller@uniklinik-freiburg.de (G.F.); florian@lampert.pro (F.L.); 2Laboratory for MEMS Applications, IMTEK—Department of Microsystems Engineering, University of Freiburg, Georges-Koehler-Allee 103, 79110 Freiburg, Germany; fritz.koch@imtek.uni-freiburg.de (F.K.); stefan.zimmermann@imtek.uni-freiburg.de (S.Z.); koltay@imtek.uni-freiburg.de (P.K.); 3Karlsruhe Institute of Technology (KIT), Institute of Functional Interfaces (IFG), Hermann-von-Helmholtz-Platz 1, 76344 Eggenstein-Leopoldshafen, Germany; sonja.haase@kit.edu (S.H.); xenia.kempter@kit.edu (X.K.); ute.schepers@kit.edu (U.S.); alisa.grimm@kit.edu (A.G.)

**Keywords:** cartilage, hydrogel, bioprinting, mesenchymal stem cells, differentiation, biocompatibility, tissue engineering

## Abstract

*Introduction:* Three-dimensional bioprinting can be considered as an advancement of the classical tissue engineering concept. For bioprinting, cells have to be dispersed in hydrogels. Recently, a novel semi-synthetic thiolene hydrogel system based on norbornene-functionalized gelatin (GelNB) and thiolated gelatin (GelS) was described that resulted in the photoclick hydrogel GelNB/GelS. In this study, we evaluated the printability and biocompatibility of this hydrogel system towards adipose-tissue-derived mesenchymal stem cells (ASCs). *Methods:* GelNB/GelS was synthesized with three different crosslinking densities (low, medium and high), resulting in different mechanical properties with moduli of elasticity between 206 Pa and 1383 Pa. These hydrogels were tested for their biocompatibility towards ASCs in terms of their viability, proliferation and differentiation. The extrusion-based bioprinting of ASCs in GelNB/GelS-high was performed to manufacture three-dimensional cubic constructs. *Results:* All three hydrogels supported the viability, proliferation and chondrogenic differentiation of ASCs to a similar extent. The adipogenic differentiation of ASCs was better supported by the softer hydrogel (GelNB/GelS-low), whereas the osteogenic differentiation was more pronounced in the harder hydrogel (GelNB/GelS-high), indicating that the differentiation fate of ASCs can be influenced via the adaption of the mechanical properties of the GelNB/GelS system. After the ex vivo chondrogenic differentiation and subcutaneous implantation of the bioprinted construct into immunocompromised mice, the production of negatively charged sulfated proteoglycans could be observed with only minimal inflammatory signs in the implanted material. *Conclusions:* Our results indicate that the GelNB/GelS hydrogels are very well suited for the bioprinting of ASCs and may represent attractive hydrogels for subsequent in vivo tissue engineering applications.

## 1. Introduction

Mesenchymal stem cells (MSCs) can be isolated from various tissues, such as bone marrow, umbilical cord and adipose tissues. They show relatively high proliferation potential in in vitro cultures, which makes it possible to expand their cell numbers rapidly for subsequent tissue engineering applications [1,2,3]. Moreover, it was shown that MSCs can differentiate in vitro, as well as in vivo, to cell types of the mesenchymal lineage, such as chondrocytes, osteoblasts and adipocytes [3,4]. In recent years, MSCs have attracted a lot of interest as very attractive cell sources for tissue engineering applications and regenerative medicine. The attractiveness is justified by the fact that these cells can be differentiated into different cell types and because MSCs can be isolated from various human tissues without severe donor site morbidity [5,6]. MSCs from various tissue sources have already been used extensively in tissue engineering applications for the generation of bone and cartilage tissues [7,8,9].

In the classical tissue engineering approach, the cells are seeded into scaffolds that are then implanted to produce a replacement tissue. In such situations, the cells are randomly distributed within the scaffold. In recent years, bioprinting techniques have been developed that can be considered to represent a progression of the classical tissue engineering concept. By using such techniques, the cells can be printed with high spatial resolution during the fabrication process to build up a replacement tissue by means of additive manufacturing. The three most-used methods in 3D bioprinting are extrusion-based bioprinting, inkjet or drop-on-demand (DoD) bioprinting and laser-assisted bioprinting [10]. For bioprinting, cells must be embedded into hydrogels, also called bioinks. These bioinks need to exhibit high cytocompatibility, must be printable and must provide sufficient mechanical stability to the printed three-dimensional construct.

In recent studies, we have tested various natural bioinks such as collagen, fibrin, gelatin and matrigel bioinks for cytocompatibility towards MSCs and endothelial cells (ECs).

These experiments revealed that most of these natural hydrogels are cytocompatible and suitable for the bioprinting of ECs and MSCs [11,12,13]. However, there are also a few shortcomings with such natural bioinks, such as the lack of consistency between hydrogels from different suppliers and different batches and the limited structural integrity of the constructs after printing. Therefore, innovative formulations of bioinks are desirable. Gelatin as a natural protein-based hydrogel is derived via the heat denaturation and hydrolysis of collagen [14]. It is FDA-approved and has a long history in tissue engineering and bioprinting [15,16]. However, due to its reversible thermal gelation and low stiffness, it can only be used to a limited extent for bioprinting applications. In the past, stimulus-sensitive gelatin hydrogels have been developed to fix these problems. For this purpose, a gelatin hydrogel was functionalized via photopolymerizable functional groups, such as methacrylamide [17]. In the presence of a photoinitiator, gelatin–methacrylamide (GelMA) can be crosslinked via photopolymerization after 3D bioprinting to make it mechanically more stable [18]. Photocurable GelMA hydrogels have already been extensively used in bioprinting applications with very good results concerning the cytocompatibility, printability and mechanical stability of the bioprinted constructs. In this context, GelMA was used for the bioprinting of bone [19], cartilage [20], cardiac patches [21], skeletal muscle [22] and other tissues [23,24].

Recently, a novel ultrafast photocurable gelatin-based hydrogel was developed particularly for bioprinting applications [25]. This thiolene hydrogel system is based on norbornene-functionalized gelatin (GelNB) and thiolated gelatin (GelS), which upon mixing result in the photoclick hydrogel GelNB/GelS. GelNB/GelS was produced in three different formulations with varying crosslinking densities (low, medium and high) [25]. While methacrylates such as GelMA polymerize in a radical chain-growth mechanism, GelNB/GelS polymerizes in a step-growth polymerization mechanism, resulting in more homogeneous networks and less toxicity to cells [26,27]. Compared to methacrylates, only a low concentration of photoinitiator is needed to initiate the reaction and the polymerization generates fewer free radicals [28,29,30].

In summary, GelNB/GelS shows several advantages over GelMA, including its superfast curing, its reduced photoinitiator requirements, its post-polymerization functionalization ability and its minimal crossreactivity with cellular components [25].

In the present study, we used three different GelNB/GelS formulations for in vitro swelling–degradation assays and for biocompatibility testing towards human adipose tissue-derived mesenchymal stem cells (ASCs) in terms of their viability, proliferation and differentiation. Because of the excellent swelling and degradation ability of the GelNB/GelS formulation with the highest crosslinking density (GelNB/GelS-high), this gel was further used for the extrusion-based bioprinting of ASCs for the production of 3-dimensional cubic constructs. Bioprinted GelNB/GelS-high constructs containing ASCs were evaluated in vivo for their cartilage formation ability and cytocompatibility in a subcutaneous mouse model.

## 2. Results

### 2.1. Storage Moduli of GelNB/GelS Hydrogels

In our initial in vitro experiments, GelNB/GelS hydrogels with varying crosslinking densities (low, medium, high) were prepared manually and cured with a UV–visible light lamp with exposure times between 140 and 160 s. In order to investigate whether the different norbornene modification levels may have an effect on the stiffness of the gels, the storage moduli were measured. As shown in Figure 1A, a clear relationship exists between the crosslinking density and the storage moduli of the gels, with GelNB/GelS- showing the lowest storage modulus (206.9 ± 25.6 Pa), GelNB/GelS-medium showing an intermediate value (581.7 ± 45.1 Pa) and GelNB/GelS-high presenting the highest storage modulus (1383.5 ± 176.3 Pa). A similar relationship between the crosslinking density and storage modulus of the hydrogels was described by Göckler et al. [25].

### 2.2. Mechanical Properties of GelNB/GelS Hydrogels

GelNB/GelS-low, GelNB/GelS-medium and GelNB/GelS-high were created in cylindrical shapes (2 mm high and 10 mm in diameter) and crosslinked and characterized in compression tests (Figure 1B). The compression tests were carried out in triplicate and compressed at a speed of 10% min^−1^. The elastic modulus was calculated from the slopes of compression stress–strain curves in the 10–20% strain area. GelNB/GelS-low was too fluid for compression tests. The maximal stress for GelNB/GelS-high was observed at 184.32 ± 39 kPa and for GelNB/GelS-medium at 172.81 ± 6.93 kPa. The elastic moduli were determined for GelNB/GelS-high and GelNB/GelS-medium as 21.53 ± 5.06 kPa and 15.56 ± 0.05 kPa, respectively; similar elastic moduli were obtained in the elastic range.

### 2.3. Viability and Proliferation of ASCs in GelNB/GelS Hydrogels

Next, we investigated whether the different storage moduli of the hydrogels may have an impact on the cell parameters of ASCs in vitro. For this purpose, ASCs were embedded in the different hydrogels and assayed for their viability, proliferation and mesenchymal lineage differentiation. For viability studies, a live–dead assay was used and representative images are shown in Figure 2A. For the quantification of the survival rates, the ratio of viable cells to total cells was determined (Figure 2B). The hydrogels showed good biocompatibility towards ASCs with viability rates between 75.7 ± 2.7% (GelNB/GelS-medium) and 81.3 ± 4.4% (GelNB/GelS-low). As shown in Figure 2C, the cell proliferation rates of ASCs embedded in the three different hydrogels were also similar and resulted in an approximately 5-fold increase in cell numbers on day 7 in relation to day 1. This led to the conclusion that GelNB/GelS hydrogels show excellent cytocompatibility towards ASCs in terms of their viability and proliferation potential.

### 2.4. Differentiation of ASCs in GelNB/GelS Hydrogels

An important feature of mesenchymal stem cells is their multilineage differentiation potential. MSCs are defined by their ability to differentiate into chondrocytes, osteoblasts and adipocytes. Therefore, we intended to investigate whether the different stiffness levels of the GelNB/GelS hydrogels may have an effect on the differentiation potential of ASCs.

For chondrogenic differentiation, ASCs were embedded in the different hydrogels and incubated for 21 days either in control medium or in chondrogenic differentiation medium. The chondrogenic differentiation was assessed via alcian blue staining (Figure 3A). As expected, the alcian blue staining increased after the incubation of ASCs in the chondrogenic differentiation medium in relation to ASCs that were incubated in the control medium. However, there was no clear difference between the various GelNB/GelS hydrogels, indicating that the differences in the stiffness levels of the gels had no effect on the chondrogenic differentiation of ASCs.

In contrast, when the ASCs were subjected to adipogenic differentiation (Figure 3B), a more pronounced lipid vesicle formation was seen in ASCs embedded in GelNB/GelS-low in relation to ASCs embedded in GelNB/GelS-medium or GelNB/GelS-high. The oil red O quantification revealed an approximately 3-fold higher lipid accumulation rate in GelNB/GelS-low in comparison to GelNB/GelS-high. As expected, the incubation of ASCs in the adipogenic differentiation medium strongly increased the lipid vesicle formation in relation to ASCs grown in the control medium, indicating that the adipogenic differentiation of ASCs can in general be achieved in all tested GelNB/GelS hydrogels.

For the osteogenic differentiation (Figure 3C), ASC-laden hydrogels were incubated in the control medium or osteogenic differentiation medium for three weeks and then analyzed for mineralization by means of alizarin red stainings. The quantification of alizarin red stainings revealed a strong induction in the mineralization of the extracellular matrix when ASC laden hydrogels were incubated in the osteogenic differentiation medium in relation to incubation in the control medium. Moreover, a direct positive correlation between the stiffness of the hydrogels and the production of the bone-like mineralized extracellular matrix could be observed. GelNB/GelS-high showed the strongest mineralization rate, followed by GelNB/GelS-medium and GelNB/GelS-low. This indicates that the osteogenic differentiation of ASCs is influenced by the mechanical properties of the hydrogel in a way that higher gel stiffness is directly associated with stronger extracellular matrix mineralization.

### 2.5. Stability of GelNB/GelS Hydrogels in an Aqueous Environment

The stability of cell-laden hydrogels in an aqueous environment is an important point, since the constructs are usually incubated in growth medium after bioprinting before they are further processed. Therefore, the different GelNB/GelS hydrogels were incubated in growth medium for up to 21 days to monitor possible swelling (weight increase) or degradation (weight decrease). For this purpose, hydrogels were weighed directly after gelation (day 0) and after different time points (Figure 4). All hydrogels showed a swelling behavior, which was most pronounced in GelNB/GelS-low, which showed a 70% increase in weight after 21 days of incubation in growth medium. In contrast, GelNB/GelS-high showed the slightest gains in weight, with only about 10% at day 17 and 23% after 21 days. An ideal hydrogel should show only minimal swelling or degradation. Therefore, we focused on the GelNB/GelS-high hydrogel, because this hydrogel showed the highest stability over the observed time course.

### 2.6. Bioprintig of ASCs in GelNB/GelS-High Hydrogel

We selected the GelNB/GelS-high hydrogel for a bioprinting approach aiming for the production of artificial cartilage tissue as a model tissue. In an initial experiment, we tested the viability of ASCs in GelNB/GelS-high printed via drop-on-demand (DoD) or extrusion method and compared this to the viability rate of ASCs dispersed manually in GelNB/GelS-high (control). Representative images of the live–dead stainings are shown in Figure 5A. The quantification of the survival rates revealed that ASCs bioprinted in GelNB/GelS-high via DoD or extrusion showed excellent viability rates of about 80%, which were not significantly different to the control group (Figure 5B).

In order to test GelNB/GelS-high for its suitability for the bioprinting of cartilage, ASCs were printed via extrusion printing to produce a cuboid structure with dimensions of 10 × 10 × 5 mm^3^. A computer-generated image of the print design can be seen in Figure 5C. A representative image of a cuboid construct directly after printing is shown in Figure 5D, indicating the good form stability of the bioprinted construct. As a negative control for further experiments, the cuboid structure was also bioprinted without ASCs (control).

After bioprinting, the constructs were incubated in vitro for 3 weeks in chondrogenic differentiation medium. Paraffin sections of bioprinted constructs were prepared and stained with a human-specific anti-vimentin antibody to visualize human ASCs or stained with alcian blue to visualize sulfated proteoglycans, which are characteristic of the cartilage-specific extracellular matrix (Figure 5E). The vimentin stainings revealed viable human ASCs in ASC-printed constructs but not in the control group. Similarly, according to the alcian blue staining, the production of sulfated proteoglycans was only seen in the ASC group but not in the control group. These results indicate that bioprinted ASCs remain viable after 3 weeks of in vitro incubation in chondrogenic differentiation medium and are able to produce sulfated proteoglycans as a component of the cartilage-specific extracellular matrix.

### 2.7. Implantation of Bioprinted Constructs

After the bioprinting and incubation of the constructs for 3 weeks in chondrogenic differentiation medium, the constructs were implanted subcutaneously into SCID mice. The constructs were retrieved after an additional 4 weeks via explantation. An example of an explanted bioprinted construct can be seen in Figure 6A. The explants were paraffin-embedded, sectioned and analyzed via vimentin, CD11b and alcian blue staining (Figure 6B). The vimentin staining revealed viable human ASCs in ASC-printed constructs but not in the control group. In the alcian blue stainings, the ASC-dependent production of proteoglycans could be observed.

Since the GelNB/GelS hydrogel is novel and had never been used before in an animal model, we were also interested in estimating the inflammatory potential of this material. Therefore, we stained the sections with an antibody against CD11b for the detection of mouse monocytes and macrophages. As shown in Figure 6B, only a very few CD11b-positive cells can be detected in the control group, as well as in the ASC group. This result demonstrates the potential of this material for in vivo tissue engineering applications.

## 3. Discussion

In this study, we investigated the suitability of a novel thiol–norbornene-functionalized gelatin hydrogel as a putative bioink for the bioprinting of human ASCs towards tissue engineering applications such as the fabrication of artificial cartilage. As previously reported, the hydrogels were produced with different amounts of norbornene, leading to various degrees of functionalization (20% for GelNB/GelS-low, 53% for GelNB/GelS-medium and 97% for GelNB/GelS-high, respectively) [25]. In our initial in vitro studies, the hydrogels were prepared manually and cured via exposure to a UV–visible light lamp for about 150 s. The UV–visible light lamp covered the complete wavelength spectrum of natural daylight. In a first set of experiments, we investigated the mechanical properties of the hydrogels via the determination of the storage moduli. As previously reported, there was a direct positive correlation between the degree of functionalization of the gels and their respective stiffness [25]. However, in the experiments performed by Gockler et al. [25], the gels were cured via long-wave UV light exposure (320–500 nm, 500 mW/cm^2^) for 1–10 s, whereas in our initial in vitro experiments, manually prepared GelNB/GelS hydrogels were cured using UV–visible light with exposure times of about 150 s. The storage moduli that were determined in our studies were consistently about 2-fold higher than that reported by Gockler et al. [25]. The observed differences in gel stiffness may be explained by the differences in the wavelength spectra of the light sources that were used to crosslink the gels, the exposure times or by differences in the manufacturing process itself. This means that fine-tuning of the mechanical properties of the GelNB/GelS hydrogels can not only be accomplished using that different degrees of functionalization of the gels, but most likely also by using different light sources and exposure times for the induction of the curing process or by using different manufacturing procedures.

The compression tests showed a wide elastic behavior between 0 and 40% strain, with similar elastic compressive moduli between approximately 10 kPa and 20 kPa for GelNB/GelS-high and GelNB/GelS-medium. With further compression up to 90% strain, increases in stiffness and plastic deformation were observed.

We were also interested in investigating whether the different mechanical properties of the GelNB/GelS hydrogels may have an impact on the cellular parameters of embedded ASCs. In this context, we observed no or minor differences between the hydrogels in the context of the viability and proliferation of ASCs. However, an impact of the stiffness of the gels on the differentiation potential of the ASCs could be observed. The hydrogel with the lowest stiffness (GelNB/GelS-low) supported the adipogenic differentiation of ASCs better than GelNB/GelS-medium or GelNB/GelS-high. In contrast, the hydrogel with the highest stiffness (GelNB/GelS-high) supported the osteogenic differentiation of ASCs better than the other two hydrogels with lower stiffness. It is known from the literature that the biomechanical properties of the environment of the MSCs play an important role in the determination of the differentiation route of the MSCs [31]. For example, it was shown that the osteogenic differentiation of MSCs is influenced by the stiffness of the surrounding extracellular matrix [32,33]. On the other hand, the adipogenic differentiation of MSCs is also regulated by the substrate elasticity and is preferred on soft substrates [34,35]. Our results also support the finding that the differentiation of MSCs is regulated at least in part by the biomechanical properties of the microenvironment. In terms of the hydrogels used in our experiments, these results indicate that the differentiation fate of ASCs can be influenced by tuning the stiffness of the GelNB/GelS hydrogels. This suggests that the bioprinting of artificial tissues can be adapted to the physiological requirements of the desired tissue.

Based on the swelling and degradation studies of the GelNB/GelS hydrogels in an aqueous environment, which demonstrated the outstanding stability of GelNB/GelS-high, we decided to focus on this particular hydrogel. In this context, we intended to validate this gel in vivo in a subcutaneous implantation model of SCID mice. As a target tissue for our bioprinting approach, we selected the cartilage according to our long term interest in the tissue engineering of osteochondral tissues. We used ASCs because these stem cells can be isolated from human adipose tissue in a minimally invasive manner [36]. Moreover, they show high proliferation potential ex vivo [37] and can be easily differentiated into chondrocytes [38,39]. ASCs were bioprinted in GelNB/GelS-high via DoD and extrusion-based printing. In both cases, the viability of the ASCs was excellent 48 h post-printing, with viability rates of about 80%. GelNB/GelS-high had already been analyzed in an extrusion-based bioprinting approach using human dermal fibroblasts with a very similar survival rate [25]. Therefore, GelNB/GelS-high represents an excellent bioink for DoD and extrusion-based bioprinting applications.

For further investigations, we bioprinted ASCs using extrusion-based printing in GelNB/GelS-high to print a cuboid structure with dimensions of 10 × 10 × 5 mm^3^. The macroscopic analysis revealed the good form stability of the bioprinted constructs and good shape fidelity. The constructs were further incubated ex vivo in chondrogenic differentiation medium for 3 weeks and then analyzed for ASC viability by means of vimentin stainings and for proteoglycan production via alcian blue staining. It became obvious that the ASCs remained viable during this incubation period and were able to produce cartilage-specific negatively charged sulfated proteoglycans.

The subcutaneous implantation of hydrogel-embedded stem cells into immunocompromised mice represents an accepted model for studying in vivo chondrogenesis [40]. Therefore, the bioprinted constructs were implanted subcutaneously into SCID mice upon the ex vivo chondrogenic differentiation of ASCs inside the constructs. After four weeks, the constructs were explanted and showed excellent dimensional stability with no signs of resorption or degradation. In these explants, the vimentin stainings revealed human-viable ASCs that were able to produce sulfated proteoglycans, as evidenced by the alcian blue stainings.

Since this hydrogel had not yet been tested in vivo, we wanted to learn about its biocompatibility in this subcutaneous implantation model. CD11b is a surface marker molecule for monocytes, macrophages and mature natural killer cells [41]. We looked for signs of inflammation using CD11b stainings in order to visualize innate immune cells that may have invaded the implants. In this context, we detected only a very few invaded inflammatory cells and there was no substantial difference between the ASC-bioprinted group and the control group where GelNB/GelS-high was bioprinted without ASCs. This result suggests that the tested hydrogel is well tolerated by the host organism. However, we detected only a very few mouse blood vessels in the explanted constructs. This offers the possibility that the low number of invaded inflammatory cells may simply be the result of the low vascularity of the constructs. Therefore, the question concerning the biocompatibility and integration properties of this new hydrogel should be further investigated in animal experiments comprising immunocompetent mice.

## 4. Materials and Methods

### 4.1. Cell Culture

Adipose-tissue-derived stem cells (ASCs) were isolated from a 39-year-old male donor according to a protocol described previously, in line with the Helsinki Declaration and approved by the ethics committee of the University Hospital Freiburg [42]. Briefly, fat tissue acquired during surgery was mechanically minced, washed with PBS and enzymatically digested at 37 °C with 2 mg/mL collagenase type II (Sigma-Aldrich, Taufkirchen, Germany, 1148090) for 3 h. Subsequently, the solution was centrifuged at 250× *g* for 10 min at room temperature and the supernatant was poured off. Thereafter, the cell pellet was resuspended in erythrocyte lysis buffer (17 mM Tris, 16 mM NH_4_Cl) for 10 min and then centrifuged again. Afterwards, the cell pellet was resuspended in PBS and filtered through a cell strainer to prevent contamination by the remaining tissue debris. Following one more centrifugation step, the cell pellet was resuspended in endothelial cell growth medium 2 (EGM-2, Lonza, Basel, Switzerland, CC3162) supplemented with 1% Pen/Strep (ThermoFisher, Dreieich, Germany, 15140122) and 10% fetal bovine serum (FBS) and seeded into a cell culture flask. The stem cells were incubated at 37 °C with 5% CO_2_ in a humidified environment and the culture medium was changed every 3 days. After expansion, the ASCs were cryopreserved and used for experiments at passages 2–5. The International Society for Cellular Therapy (ISCT) has proposed a set of standards to define MSCs [43]. MSCs must be able to adhere to plastic and must show a fibroblast-like morphology. Moreover, these cells must be able to differentiate into osteoblasts, adipocytes and chondrocytes ex vivo and must show a typical expression pattern of cell surface molecules (CD105^+^, CD90^+^, CD73^+^, CD45^−^, CD14^−^, CD34^−^, HLA-DR^−^). We were able to identify the isolated ASCs as mesenchymal stem cells according to these standards.

### 4.2. Preparation of Hydrogels

The synthesis of norbornene-functionalized gelatin (GelNB) and thiolated gelatin (GelS) was previously described [25]. GelNB/GelS is a semi-synthetic hydrogel that was produced in three different degrees of functionalization to yield materials with different crosslinking densities and levels of mechanical stiffness (low, medium and high). Depending on the desired crosslinking density and degree of functionalization (DoF), GelNB and GelS were used with the corresponding number of equivalents. The 10% (*w*/*v*) GelNB and 5% (*w*/*v*) GelS were dissolved in sterile PBS at 70 °C. The further use of the terms “GelNB” and “GelS” refers to 10% and 5% solutions in PBS, respectively. The photoinitiator lithium phenyl(2,4,6-trimethylbenzoyl)phosphinate (LAP) was dissolved in DMSO at 37 °C for 15 min at a concentration of 10% and stored at −80 °C. With the exception of GelNB/GelS-low, the solutions of GelNB and GelS were mixed to ensure equimolar amounts of both functional groups in a 5% (*w*/*v*) hydrogel precursor solution.

The composition of GelNB/GelS-low was based on 49.5% (*v*/*v*) of a 10% GelNB (2 equivalents, DoF 53%) stock solution, 49.5% (*v*/*v*) of a 5% GelS (1 equivalent, DoF 20%) stock solution and 1% (*v*/*v*) of a 10% photoinitiator LAP stock solution, resulting in a hydrogel precursor solution of 7.5%. GelNB/GelS-medium consisted of 25% (*v*/*v*) of GelNB (2 equivalents, DoF 53%), 50% (*v*/*v*) of GelS (5 equivalents, DoF 50%), 24.7% (*v*/*v*) of the Dulbecco’s modified Eagle’s medium (DMEM, Thermo Fischer, 41965039) and 0.3% (*v*/*v*) of a 10% LAP stock solution. GelNB/GelS-high consisted of 17.3% (*v*/*v*) of GelNB (10 equivalents, DoF 97%), 65.5% (*v*/*v*) of GelS (5 equivalents, DoF 50%), 16.9% (*v*/*v*) of DMEM and 0.3% (*v*/*v*) of the 10% LAP stock solution.

After the individual components had been mixed, 500 µL of each hydrogel was pipetted into surgical steel cylinders (12 mm, Crazy Factory, DFLTU). The hydrogels were photopolymerized with a 300 W lamp (Osram Ultra-Vitalux E27, München, Germany), which had a radiation spectrum similar to that of daylight. During the exposure, a distance of 150 mm between the hydrogel and the lamp was maintained. The exposure time was 160 s for GelNB/GelS-low, 140 s for GelNB/GelS-medium and 150 s for GelNB/GelS-high. After the photopolymerization of the hydrogels, the surgical steel cylinders were carefully removed using a scalpel and the samples were covered by cell culture medium in a 12-well plate.

### 4.3. Swelling and Degradation Assays

To evaluate the possible swelling or degradation of hydrogels in an aqueous environment, the mass swelling ratios were determined. After obtaining blank measurements of each specific dish, 500 µL of each hydrogel was plated in tissue culture dishes (ThermoFisher, Dreieich, Germany) and crosslinking was induced via photopolymerization using a 300 W lamp, as described above. The gels were incubated in EGM-2 with 10% FCS and 1% Pen/Strep at 37 °C for up to 21 days. The measurements were performed in triplicate directly after photopolymerization of the hydrogels (*W*_0_) and then after the addition of growth medium on days 1, 4, 7, 11, 14, 17 and 21 (*W* swollen). Prior to each measurement, the entire supernatant was removed from the dish to allow the weight measurement of the remaining hydrogel. Blank measurements of the dishes were subtracted. After every weighing procedure, every sample was covered by fresh culture medium. The mass swelling ratio was determined using the following formula:$$\mathrm{Mass}\;\mathrm{swelling}\;\mathrm{ratio}=\frac{W\;\mathrm{swollen}-W_{0}}{W_{0}}\times 100\%\;$$

### 4.4. Determination of the Storage Modulus

To determine the storage modulus (G′, (Pa)), the hydrogels were plated into a mold with a diameter of 25 mm and a height of 200 µm, photopolymerized with a 300 W lamp as described above and incubated overnight in PBS at 37 °C. The storage modulus was determined as the mean value of at least 153 measuring points at 37 °C using an MCR 101 rotational rheometer from Anton Paar with a plate measuring 25 mm in diameter. All measurements were performed at 1 1/s after a frequency sweep from 0.1 to 100 1/s to ensure the measurements were within the plateau regime.

### 4.5. Determination of the Mechanical Properties of the Hydrogels

The mechanical properties of the hydrogels were investigated using an AGS-X Series Universal Electromechanical Test Frame (Shimadzu Inc., Kyoto, Japan). Cylindrical samples (2 mm high and 10 mm in diameter, photopolymerized) were used for compression tests at a speed of 10% min^−1^. Three samples were prepared for each compression experiment. The elastic modulus was calculated from the slope of compression stress–strain curves in the 10–20% strain area.

### 4.6. Cell Viability Assay

A Live/Dead Viability/Cytotoxicity Kit (ThermoFisher, Dreieich, Germany, L3224) was used to investigate the influence of the hydrogels on the ASCs’ viability. According to the supplier’s instructions, 5 × 10^4^ ASCs were mixed with 500 µL of hydrogel, photopolymerized as described above, covered by EGM-2 with 10% FBS and 1% Pen/Strep and incubated in a 12-well plate for 48 h. After the removal of the culture medium, the triplicates were washed once with PBS and the freshly prepared live–dead solution was added to each sample. Subsequently, the samples were incubated for 30 min at room temperature before the images were captured using an Observer Z1 microscope (Carl Zeiss, Oberkochen, Germany). Ten regions of interest in each well were evaluated. To quantify the numbers of living and dead cells, ImageJ software (version 1.47) was used. The survival rate was calculated as the ratio of viable cells to total cells.

### 4.7. Cell Proliferation Assay

To determine the influence of the hydrogels on the ASCs’ proliferation, 1 × 10^5^ cells were mixed into the hydrogels before photopolymerization and incubated over 7 days in EGM-2, 10% FCS and 1% Pen/Strep. The cell activity was measured using the cell activity assay CellTiter96 Aqueous One Solution (Promega, Walldorf, Germany, G3580). This assay is based on a colorimetric reaction in which the tetrazolium salt MTS (3-(4,5-dimethylthiazol-2-yl)-5-(3-carboxymethoxyphenyl)-2-(4-sulfophenyl)-2H-tetrazolium) is reduced by living cells to formazan, resulting in a color change of the medium. Triplicates of each hydrogel were plated in a 12-well plate (500 µL per well) for measurements on days 1, 3 and 7. The media were changed on days 3 and 5. Prior to adding the dye, all samples were washed twice with PBS and the media were renewed. After incubation for 3 h at 37 °C, the optical density (OD) was measured using a SpectraFluor Plus colometric reader (Tecan, Männedorf, Switzerland) at 490 nm. To determine the changes in cell activity, blank measurements with hydrogels without cells were carried out. These values were substracted from the experimental values. The changes in OD were normalized to day 1 for each hydrogel. The OD values at day 1 were set to 100%.

### 4.8. Differentiation of the Encapsulated ASCs

For the differentiation experiments, 1 × 10^6^ ASCs/mL were mixed with 500 µL of hydrogel. After photopolymerization, the cell-laden hydrogels were incubated for 21 days at 37 °C. Six samples were fabricated for each differentiation path (osteogenic, chondrogenic, adipogenic) and each hydrogel. Three of them were stimulated with the differentiation medium and another three with the control medium. The osteogenic medium was based on low-glucose DMEM (Thermo Fisher, 31885023) supplemented with 10 mM ß-glycerophosphate (Sigma, G9422), 0.1 µM dexamethasone (Sigma, D4902), 50 µM ascorbic acid (Sigma, BP461), 10% FBS and 1% Pen/Strep. The control medium for this group contained only DMEM low glucose, 10% FBS and 1% Pen/Strep. For the chondrogenic differentiation medium we used high-glucose DMEM (Thermo Fisher, 41965039) supplemented with 0.1 µM dexamethasone, 50 µM ascorbic acid, 50 µM L-proline (Sigma, P0380), 1% (*v*/*v*) insulin–transferrin–sodium selenite (ITS + 1, Sigma, I2521), 10 ng/mL TGF-β3 (Sigma, SRP3171) and 1% Pen/Strep. The control medium in this group consisted of high-glucose DMEM and 1% Pen/Strep. The adipogenic differentiation medium was based on DMEM/Hams F-12 (Thermo Fisher, 31331028), 1 µM insulin (Sigma, I6634), 0.25 mM 3-Isobutyl-1-methylxanthine (IBMX, Sigma, I7018), 200 µM indomethacin (Sigma, I7378), 1 µM dexamethasone, 3% FBS and 1% Pen/Strep. For this group, DMEM/Hams F-12, 3% FBS and 1% Pen/Strep were used as the control medium.

### 4.9. Alizarin Red Staining and Quantification

After 21 days of incubation, the osteogenic-stimulated and respective control constructs were cut in half using a scalpel. One half was fixed in 4% formalin for 10 min, dehydrated overnight using a tissue processor (Leica Biosystems, Nußloch, Germany, TP1020) and embedded in paraffin. The paraffin samples were cut into slices with a thickness of 7 µm. Alizarin red staining was performed to assess the calcification of the extracellular matrix in the osteogenic-stimulated constructs. The 2% alizarin red (Sigma, A5533) solution was freshly prepared in H_2_O bidest. After the pH was adjusted to 4.2 with 10% ammonium hydroxide, the solution was filtered through Whatman paper. The deparaffinized (via descending ethanol series (100–96–96–70%)) and rehydrated sections were washed twice with PBS and stained in alizarin red solution for 2 min. Afterwards, the stained sections were washed several times with PBS and air-dried. Finally, the slides were briefly dipped in Xylol (Fisher Chemical, Schwerte, Germany, X025015) and mounted with Entellan (Merck, Darmstadt, Germany, 107961).

For the quantification of the mineralization, half of each construct was used. These were fixed with 4% formalin for 30 min and washed thoroughly with H_2_O bidest. Afterwards, the constructs were stained with a filtered 2% alizarin red solution for five minutes while shaking. To remove the excess dye from the hydrogels, we washed the constructs in a 6-well plate with PBS over 16 h. The PBS was changed six times. Following a drying step for 30 min at 37 °C, the constructs were weighed, transferred into reaction tubes and mechanically disintegrated using a micro-pestle (Eppendorf, Hamburg, Germany, 0030120973). After adding a 10% cetylpyridinium chloride (CPC) solution (0.1 µM, pH 7.0), the samples were incubated for 2 h at 37 °C while shaking (600 rpm). Subsequently, the samples were centrifuged at 20,000 rpm for 15 min and the supernatants were transferred onto a 96-well plate. The optical density was measured using a Multifunction Multiplate Reader (Tecan, Infinite M200) at 492 nm. The determined results were normalized to the respective weight of each sample.

### 4.10. Alcian Blue Staining

Alcian blue staining is used to detect negatively charged sulfated proteoglycans in a cartilage matrix or in chondrogenic differentiated cultures [44]. The 1% alcian blue (Sigma, 10909) solution was prepared in 3% acetic acid and the 0.5% nuclear fast red (Roth, Karlsruhe, Germany, 60760) solution was prepared in H_2_O bidest. The constructs were fixed in 4% formalin for 10 min, dehydrated overnight using a tissue processor (Leica Biosystems, TP1020) and embedded in paraffin. The paraffin samples were cut to 7 µm in thickness. The sections were deparaffinized, rehydrated and immersed in 3% acetic acid for 3 min. Afterwards, the slides were stained with freshly filtered alcian blue solution (pH 1.0) for 30 min and dipped briefly in 3% acetic acid. Following a washing step under cold running tap water for 5 min, the sections were counterstained with freshly filtered nuclear fast red solution for 1 min. The excess dye was removed by repeatedly and gently washing them with H_2_O bidest. Subsequently, the slides were dipped in Xylol and mounted with Entellan.

### 4.11. Oil Red O Staining and Quantification

To demonstrate the adipogenic differentiation of ASCs in hydrogels, oil red O staining was performed. This dye stains intracellular neutral lipids (such as triglyceride) and thus the fat vacuoles red. Oil red stock solution was prepared by dissolving 0.5 g of oil red O (Sigma, O-0625) in 100 ml of H_2_O bidest. The working solution was composed of two parts of H_2_O bidest and three parts of the stock solution. The hydrogels were fixed with 4% formalin for 10 min at room temperature, washed thoroughly with PBS and overlaid with 60% isopropanol for 5 min. After the isopropanol was poured off, a freshly filtered oil red working solution was applied to the constructs for 15 min at room temperature. Following the washing steps with PBS (until the supernatant became clear), the stained constructs were observed using an Observer Z1 microscope (Carl Zeiss, Germany) and images of random areas were captured.

A digital image analysis (DIA) was carried out to quantify the intracellular accumulation of the oil red dye in the stained samples. This method had already been validated and used by other research groups [45,46,47]. For this purpose, 54 randomly picked microscopic images were evaluated using Image J software. All images had the same magnification (200×) and the same resolution (300 dpi). Due to the three-dimensionally arranged cells in the hydrogel, the light conditions of the background were not the same for each exposure. For this reason, each image had to be analyzed manually and adjusted to the background using threshold settings. This ensured that the red pixels were only counted intracellularly (in fat vacuoles). To compare the adipogenic differentiation of the ASCs in hydrogels, the red pixels within the cells were counted and divided by the total number of cells on the respective image.

### 4.12. Bioprinting of ASCs

#### 4.12.1. Bioprinting for the Evaluation of the Viability of ASCs

To investigate the influence of the printing process on the survival of ASCs, the GelNB/GelS-high hydrogel containing 2 × 10^6^ ASCs/mL was printed by extrusion or by drop-on-demand (DoD). For extrusion-based bioprinting, we used a 2 mL syringe (Injekt Luer Lock, B Braun, Melsungen, Germany, 4606701V) that was actuated by a piston-driven syringe pump. The meandering lines were formed by a Teflon-coated metal nozzle with a 250 µm inner diameter (Globaco, TF720100PK). The flow rate was set to 5 µL/s and the printing speed to 4 mm/s. For DoD bioprinting we used a piezo-driven dispenser (PipeJet P9, Biofluidix, Freiburg, Germany), which we had already used for different bioprinting applications [11,12,13]. The dispenser was connected to a reservoir via a silicone tube and exhibited a straight nozzle with an inner diameter of 500 µm. The stroke velocity was adjusted to 140 µm/s and the stroke size was adjusted to 35 µm with a feed rate of 4 mm/s at a dispensing frequency of 10 Hz. In both cases meandering lines were printed in a square form of 1 cm × 1 cm in triplicate in a 6-well plate. As a control, ASC-containing GelNB/GelS-high was also pipetted manually using a 1 mL pipette (Eppendorf). Thereafter, the hydrogels were crosslinked with UV light at 365 nm for 20 s. The samples were incubated for 48 h at 37 °C in EGM-2 with 10% FBS and 1% Pen/Strep. The cell viability was analyzed using the Live/Dead Viability Kit (Life Technologies, L3224).

#### 4.12.2. Bioprinting of ASCs for Implantation

Extrusion-based bioprinting was performed on a three-axis bioprinter prototype with heating and cooling control for the extruder and substrate (Biofluidix, Germany). For the in vivo evaluation, constructs with dimensions of 10 × 10 × 5 mm^3^ were designed and printed with a Teflon-coated metal nozzle with an inner diameter of 250 µm using disposable 2.5 mL syringes as a reservoir (B. Braun, Melsungen, Germany). A total of 12 constructs divided into 2 groups (GelNB/GelS-high without ASCs and GelNB/GelS-high with 4 × 10^6^ ASCs/mL) were printed. The constructs were printed at 18 °C and 10 °C as the substrate temperatures. The constructs were printed in 10 layers with a height of 500 µm each. The GelS/GelN-high hydrogel was crosslinked via exposure to UV light at 365 nm every second layer for 20 s. The printing time for one construct was around 10 min. After bioprinting, the constructs were incubated in chondrogenic differentiation medium (DMEM high glucose, 0.1 µM dexamethasone, 50 µM ascorbic acid, 50 µM L-proline, 1% (*v*/*v*) Insulin-transferrin-sodium selenite (ITS + 1), 10 ng/ml TGF-β3 and 1% Pen/Strep) for 3 weeks at 37 °C with 5% CO_2_ in a humidified atmosphere. The medium was changed twice per week.

### 4.13. Immunohistochemical Analysis

Constructs were fixed in 4% formalin for 10 min, dehydrated overnight using a tissue processor (Leica Biosystems, TP1020) and embedded in paraffin. The paraffin samples were cut to 5 µm in thickness. The sections were deparaffinized and rehydrated. Antigen retrieval was performed by boiling the slides in citrate buffer. After the slides were transformed to the humid chamber, the endogenous peroxidase activity was blocked by incubating slides in 3% H_2_O_2_ for 20 min. Following a washing step with H_2_O bidest, nonspecific binding was blocked using 5% goat serum (Dako, Frankfurt, Germany, X0907) diluted in H_2_O bidest for the subsequent vimentin stainings or 5% donkey serum for the CD11b stainings. Afterwards, the samples were washed three times with TBST (150 mM, NaCl 50 mM Tris, 0.05% Tween 20, pH 8.2), and the primary antibody mouse anti-human vimentin (1:100 in 1% goat serum, Dako, M0725) or rabbit monoclonal CD11b (1:5000, abcam, Berlin, Germany, ab133357) in PBS was applied and incubated overnight at 4 °C. For the negative control, 1% goat serum in H_2_O bidest or only PBS was used instead of the primary antibodies. In the next step, all samples were washed several times with TBST and secondary goat anti-mouse antibodies (ready to use, Dako, K4001) or donkey anti-rabbit antibodies (1:100 in PBS, GE Healthcare, Freiburg, Germany, NA934) were applied for 45 min at room temperature. After the slides were washed three times with TBST, 3,3′-diaminobenzidine (DAB; Vector Laboratories, Eching, Germany, SK4100) staining was performed in accordance with the supplier’s instructions. The staining reactions were incubated for 4 to 7 min at room temperature. Following a washing step with PBS, the slides were counterstained with hematoxylin, dehydrated and finally permanently mounted with Entellan. The images were taken with an Axio Imager M2 microscope (Carl Zeiss).

### 4.14. Animal Experiments

The bioprinted and chondrogenically differentiated constructs were implanted into the dorsal subcutaneous pouches of 4–-week-old SCID mice (C.B.-17-SCID, Charles River, Sulzfeld, Germany). The German regulations for the care and use of laboratory animals were met at all times. All experiments were approved by the animal care committee of the University of Freiburg. The animals were housed in the veterinary care facility of the University of Freiburg Medical Center. The animals were randomly assigned to one of the experimental groups (control: bioprinted constructs without ASCs; GelNB/GelS-high + ASCs: bioprinted constructs with ASCs). Three animals were used for each group and two constructs were implanted in each animal; one left and one right of the midline. Thus, each group consisted of six constructs (n = 6). The implantation of the constructs was carried out under general anesthesia via the inhalation of 5% isoflurane and maintained via the inhalation of 1.5% isoflurane. After dorsal skin disinfection, a full-thickness incision was made to create a subcutaneous pouch (approximately 2 cm^3^) on each side of the spine to accommodate the construct. Subsequently, the bioprinted constructs were implanted into the respective pouch. Finally, the wound was closed with interrupted 6-0 vicryl sutures. Four weeks after implantation, the mice were sacrificed and the constructs were explanted for evaluation.

### 4.15. Statistical Analysis

Statistically significant differences between the sample groups were determined using an unpaired Student’s *t*-test. Statistical significance was defined when *p* < 0.05.

## 5. Conclusions

In summary, our experiments have shown that the GelNB/GelS hydrogel system is very suitable as a bioink for embedding and bioprinting ASCs for prospective tissue engineering applications. The viability, proliferation and differentiation potential of ASCs is supported by the GelNB/GelS hydrogels. Through the modification of the degree of functionalization, it is possible to fabricate GelNB/GelS hydrogels of varying stiffness levels. Remarkably, we observed that a softer hydrogel (GelNB/GelS-low) supported the adipogenic differentiation of ASCs, whereas a harder hydrogel (GelNB/GelS-high) supported their osteogenic differentiation. This indicates that the differentiation fate of ASCs can be influenced via the mechanical properties of the GelNB/GelS hydrogel. Overall, we were able to show that GelNB/GelS-high represents an excellent hydrogel for the bioprinting of ASCs and may also be attractive for subsequent tissue engineering applications in vivo.

## Figures and Tables

**Figure 1 ijms-23-07939-f001:**
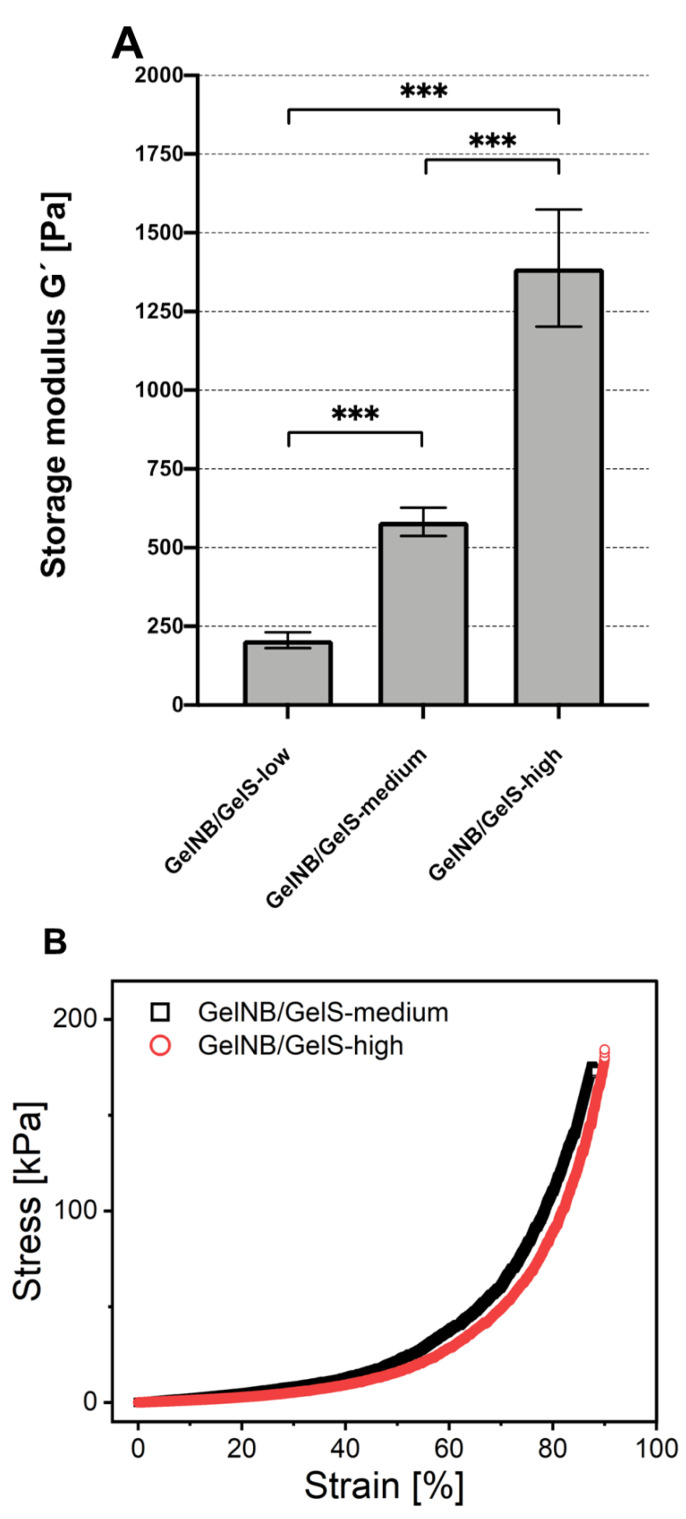
Determination of the storage moduli and elastic moduli of GelNB/GelS-low, GelNB/GelS-medium and GelNB/GelS-high hydrogels. The tests were performed for crosslinked hydrogels at RT. (**A**) For the determination of the storage modulus, all dynamic moduli were measured at a 1 Hz oscillation frequency in the plateau regime. Means ± SD are shown for at least 153 measuring points of one hydrogel per experimental group. G′: storage modulus (Pa). Asterisks indicate statistically significant differences between the groups. (*** *p* < 0.0005). (**B**) Representative compression stress–strain curves of GelNB/GelS-medium and GelNG/GelS-high for the determination of the elastic modulus. Samples (n = 3) were compressed at a speed of 10% min^−1^. The elastic modulus was calculated from the slopes of the compression stress–strain curves in the 10–20% strain area.

**Figure 2 ijms-23-07939-f002:**
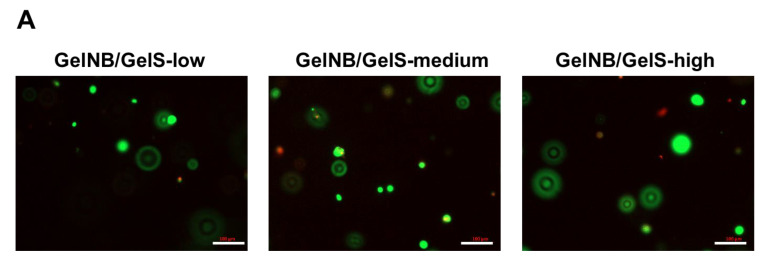
Viability and proliferation testing. ASCs were mixed into the three GelNB/GelS hydrogels. (**A**) Viability testing was performed using a live–dead assay and representative images are shown. Scale bars: 100 µm. (**B**) To determine the survival rates, 3 × 10 regions of interest (ROI) were evaluated. The ratio of viable cells and total cells provided the survival rates. Means ± SD are shown (n = 3). (**C**) Proliferation of ASCs was measured using a MTS assay. Based on measurements, cell proliferation rates are expressed as percentages of day 1. Means ± SD are shown (n = 3). Asterisks indicate statistically significant differences between the groups in relation to the GelNB/GelS-medium group (* *p* < 0.05, ** *p* < 0.005).

**Figure 3 ijms-23-07939-f003:**
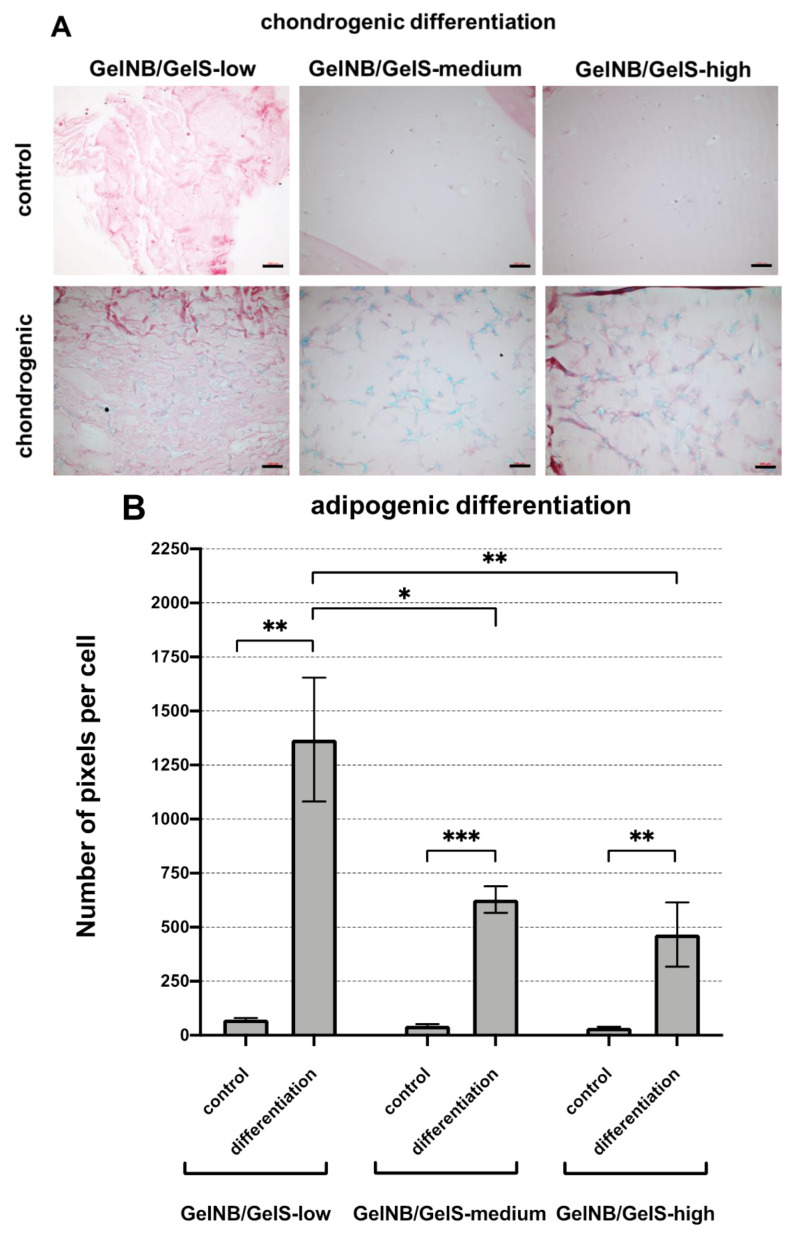
Chondrogenic, adipogenic and osteogenic differentiation of ASCs in GelNB/GelS-low, GelNB/GelS-medium and GelNB/GelS-high hydrogels. ASCs were dispersed in the hydrogels in triplicate and grown in normal growth medium (control) or the respective differentiation media for 21 days. (**A**) Chondrogenic differentiation was assessed via alcian blue staining. Shown are represenative images from ASCs grown in triplicate in each group (n = 3). Scale bars: 100 µm. (**B**) Adipogenic differentiation was assessed by oil red O staining and the quantification of lipid vesicle formation was performed via digital image analysis (n = 3). Values represent the average numbers of red-stained pixels per cell. Shown are means ± SD (n = 3). (**C**) Osteogenic differentiation was assessed via alizarin red staining and the quantification of mineralization was achieved via the extraction of alizarin red and photometric measurements. Shown are means ± SD (n = 3). Asterisks indicate statistically significant differences between the groups (* *p* < 0.05, ** *p* < 0.005, *** *p* < 0.0005).

**Figure 4 ijms-23-07939-f004:**
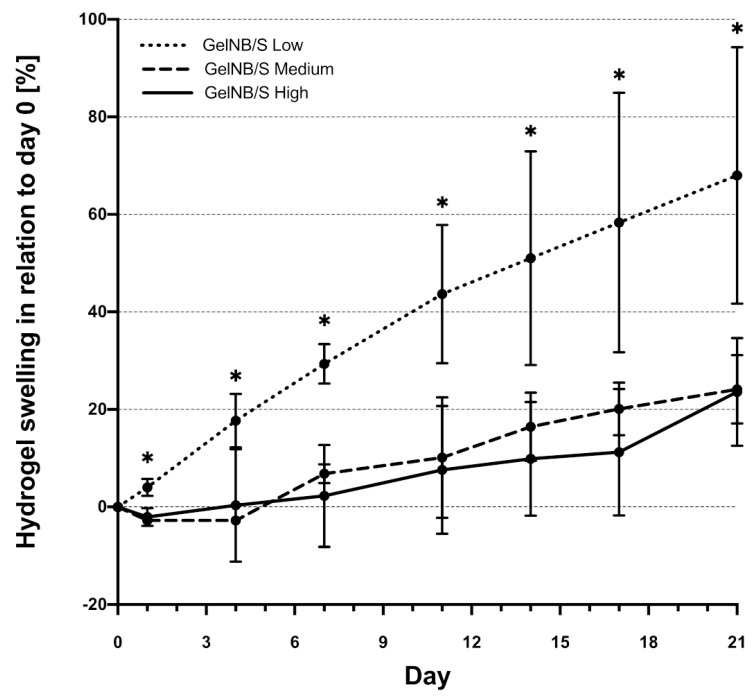
Swelling or degradation properties of GelNB/GelS-low, GelNB/GelS-medium and GelNB/GelS-high hydrogels. Hydrogels at different concentrations were plated in triplicate in tissue culture dishes and after gelation were overlaid with growth medium. The weights of the gels after the removal of the growth medium was determined at days 0, 1, 4, 7, 11, 14, 17 and 21. Swelling is indicated by an increase in weight, whereas degradation of the hydrogels is indicated by a decrease in weight. Values at the respective days were normalized to day 0 and are expressed as percent swelling values in relation to day 0. Shown are means ± SD (n = 3). Asterisks indicate statistically significant differences between the groups in relation to GelNB/GelS-high (* *p* < 0.05).

**Figure 5 ijms-23-07939-f005:**
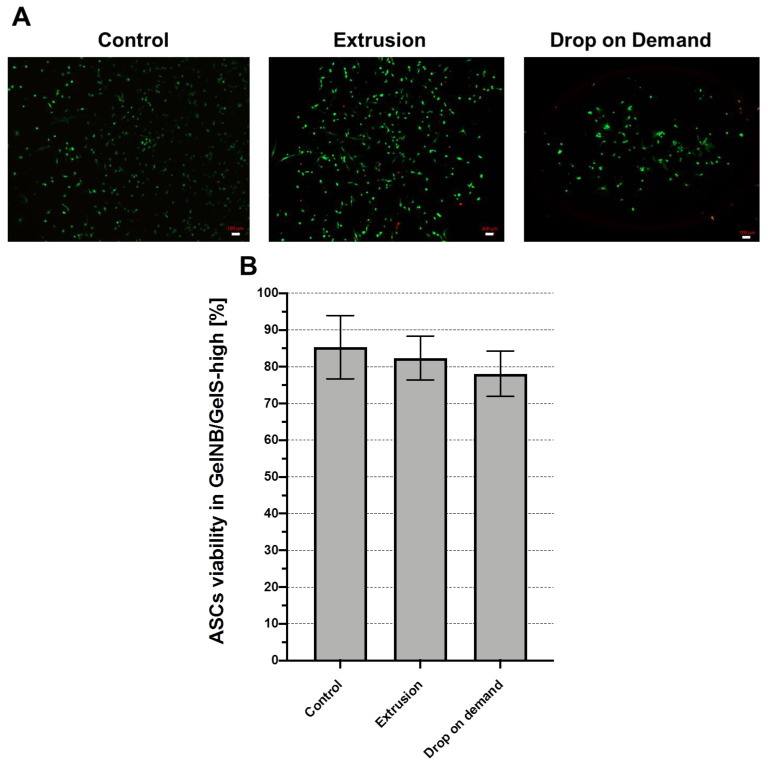
Bioprinting of ASCs in GelNB/GelS-high hydrogel. (**A**) The viability of ASCs was determined 48 h after extrusion-based bioprinting or DoD bioprinting and after manual dispersion of ASCs in GelNB/GelS-high (control) using a live–dead assay. Representative images images from ASCs grown in triplicate in each group are shown (n = 3). Scale bars: 100 µm. (**B**) Quantitative determination of the survival rates via the calculation of the ratio of viable cells to total cells. Shown are means ± SD (n = 3). (**C**) Computer-generated image of the print design for the extrusion-based bioprinting of ASCs in GelNB/GelS-high for implantation. (**D**) Macroscopic image of a representative construct directly after bioprinting. (**E**) Constructs, bioprinted with or without ASCs (control), were incubated in vitro for 3 weeks in chondrogenic differentiation medium. Constructs were paraffin-embedded, sectioned and alcian-blue- or anti-vimentin-stained. Scale bars: 100 µm.

**Figure 6 ijms-23-07939-f006:**
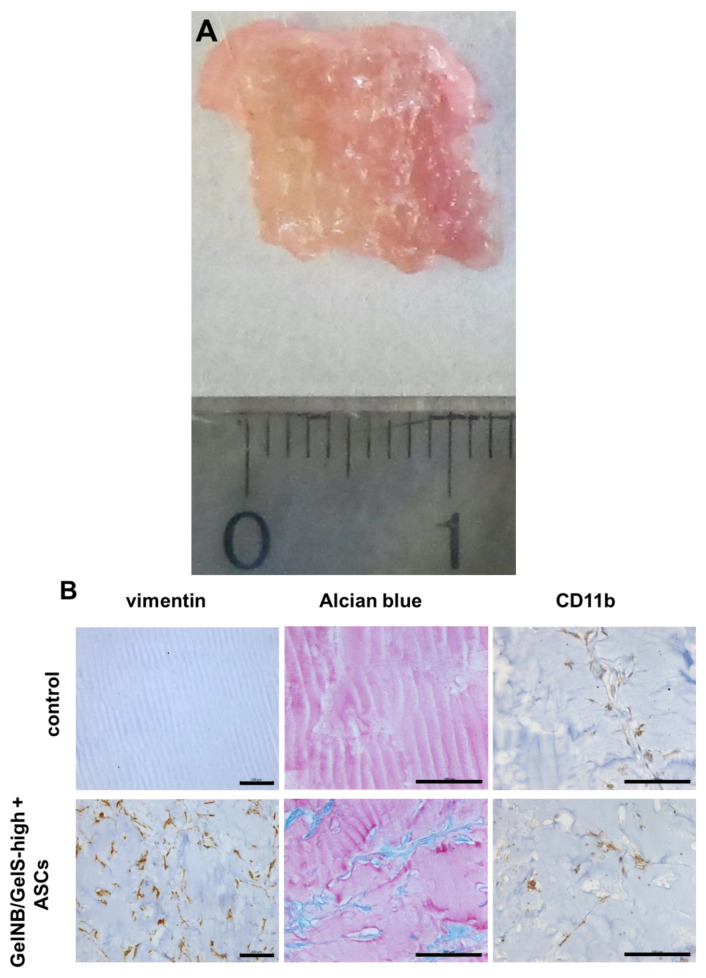
Constructs were implanted subcutaneously into SCID mice and were explanted after 4 weeks. (**A**) Macroscopic image of an explanted ASC-printed construct. (**B**) Paraffin sections of explants were stained with a human-specific antibody against vimentin to visualize ASCs or with an anti-CD11b antibody for the visualization of mouse innate immune cells. The alcian blue staining of paraffin sections was performed to visualize sulfated proteoglycans. Representative images images from each group (n = 6) are shown. Scale bars: 100 µm.

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
