# Peer review of "Evaluation of a Novel Thiol–Norbornene-Functionalized Gelatin Hydrogel for Bioprinting of Mesenchymal Stem Cells"

_ijms, 2022, doi:10.3390/ijms23147939_

Round 1

Reviewer 1 Report

  • Introduction. The novelty and the relevance of the work is unclear. What is the unique advantage of the proposed approach over the others (GelMA bioinks, FRESH technique or concentrated collagen bioinks)? I would also recommend to move the text "43 The International Society for Cellular Therapy...48 CD34-, HLA-DR-) to the Methods.
  • Results. Fig 4A. I can't see a big difference between groups in persented picture. How does this compare with the result indicated by the authors "3-fold higher lipid accumulation in GelNB/GelS-low compared to high"?
  • Results. Fig 5A. Same question as for Fig 4A
  • Results. Bioprinting of constructs for implantation. What about viability of the cells in constructs after 3 weeks of cultivation? what about shape changing of this constructs, because authors it this paper also showed the 23% of swelling for GelNB/GelS-high. Can swelling effect on the viability?
  • Results. Fig 7D. It doesn't look like as good printing fidelity. What dimensions of the resulting construct? Authors should quantify it.
  • Results. Implantation. It should be provided a histological analysis of the scaffold with before implantation, since is essential to establish what was the starting point, especially after 3 weeks in culture.
  • Results. Implantation. If authors want to discuss about biocompatibility of the scaffold, the should provide H&E staining of surrounding tissues.
  • Results. Implantation. The lack of integration of the implant with the surrounding tissues 4 weeks after implantation indicates the inertness of the material, and not its biocompatibility. Lack of blood vessels in the scaffold will lead to cell death inside the scaffold. This result is a good highlight of main problems which may appear in vivo after modification of biomaterials via chemical or photochemical crosslinking.
  • The statement that authors achieved cartilage formation in vivo does not have a substantial evidence base. "a cartilage-like extracellular matrix" in unclear
  • Discussion and Conclusion mus't be modified after making changes in accordance with the comments above

Reviewer 2 Report

In this manuscript (ijms-1663339), the authors have 3D bioprinted constructs of thiol-norbornene functionalized gelatin hydrogels and evaluated the biocompatibility well. This study is interesting and can be considered for publication after a minor revision.

  1. The printability of the developed hydrogel systems was not evaluated well. No one can see the 3D printed structure (post-printing fidelity or mesh network) as per the designed model. Is not that important? If yes, then the authors could prepare only simple hydrogels with cells for evaluating the biocompatibility. 3D bioprinting is very important while fabricating spatial control over the structure.
  2. The authors should provide the digital images of 3D printed constructs with or without cells for a clear understanding in the case of the printing process.
  3. Rheological properties are also very important for printing behavior (i.e. flow, extrusion). Therefore, actual plots (e.g., storage modulus vs frequency sweep) should be provided for a clear understanding of the behavior of hydrogels.

Round 2

Reviewer 1 Report

I appreciate authors attempt for answering of my questions, but I still have got some of them unclear

  1. Fig. 5D. I can agree that the fig. 5D indicated form stability. However, due to the lack of evaluation methods of printing fidelity, it is impossible to infer any degree of printing fidelity.
  1. Results. Implantation. The authors provided evidence that the cells inside the printed construct retained their viability and functionality after implantation. This is a significant result, demonstrating that this material is biocompatible with cells inside. However, there are standard procedures for evaluating the biocompatibility of a material (especially for new material). I would recommend that the authors remove statement 555 "suggesting good in vivo biocompatibility of this hydrogel" or change the wording. For example, "this result demonstrates the potential of this material for in vivo tissue engineering applications»
  1. Regarding authors comment on my comment about lack of integration of the biomaterial with surrounding tissues «We do not think that the material is completely inert, because we have seen adhesions between the implanted material and the implant bed of the mouse, making explantation sometimes mechanically difficult».                                                                                                                                                                   It doesn't follow from the data you've provided. Figure 6a shows an almost unchanged material, in which animal tissues are not visible. The fact that you were able to separate the material from animal tissues after 4 weeks of subcutaneous implantation raises many questions about the biocompatibility of this material. In particular,                                                                                                                                                - How did the host tissue react to implantation?                                                                                                  - What  about the presence/absence of giant cells of forreign bodies on the surfface of the implant or in interphase zone?                                                                                                                                               - What about the migration of host cells (like, fibroblasts) into this implant in both groups?                              These questions are for thought provoking and do not require an immediate answer. Good luck with ongoing publications

Author Response

Point-by-point reply to the comments of reviewer-1, round-2

In the second round of this review process, we got three minor comments from one of the reviewers.

In response to the comments, we revised the manuscript by modifying the text in agreement with the valuable considerations. All changes in the text of the revised manuscript have been done in the Track changes modus of the word program.

Reviewer: 1

Comments to Author(s)

  • 5D. I can agree that the fig. 5D indicated form stability. However, due to the lack of evaluation methods of printing fidelity, it is impossible to infer any degree of printing fidelity.

We thank the reviewer for this valuable comment and agree with him that form stability of the bioprinted construct can be detected, which, however, does not necessarily indicate high printing accuracy. Therefore, we decided to remove the statement concerning the printing fidelity in all sections of the manuscript.

  • Implantation. The authors provided evidence that the cells inside the printed construct retained their viability and functionality after implantation. This is a significant result, demonstrating that this material is biocompatible with cells inside. However, there are standard procedures for evaluating the biocompatibility of a material (especially for new material). I would recommend that the authors remove statement 555 "suggesting good in vivo biocompatibility of this hydrogel" or change the wording. For example, "this result demonstrates the potential of this material for in vivo tissue engineering applications»

According to the comment of the reviewer, we have changed the wording and have replaced the part concerning in vivo compatibility by the sentence suggested by the reviewer.

  • Regarding authors comment on my comment about lack of integration of the biomaterial with surrounding tissues «We do not think that the material is completely inert, because we have seen adhesions between the implanted material and the implant bed of the mouse, making explantation sometimes mechanically difficult».                                                                                                                                                       It doesn't follow from the data you've provided. Figure 6a shows an almost unchanged material, in which animal tissues are not visible. The fact that you were able to separate the material from animal tissues after 4 weeks of subcutaneous implantation raises many questions about the biocompatibility of this material. In particular, How did the host tissue react to implantation?                                                                                                

- What  about the presence/absence of giant cells of forreign bodies on the surfface of the implant or in interphase zone?

- What about the migration of host cells (like, fibroblasts) into this implant in both groups?                              

These questions are for thought provoking and do not require an immediate answer. Good luck with ongoing publications

We thank the reviewer for these valuable suggestions. Indeed, the interaction between the novel hydrogel and the host organism needs further investigations. We have been able to show that the biomaterial is biocompatible towards the ASCs but for investigating the materials properties in the interaction with the host organism, more animal experiments, preferably with immunocompetent mice, will we necessary. We will address these questions in further experiments in the future.

We hope that we have taken care of the suggestions to improve the manuscript and thank again for the very cooperative review process, which helped a lot to improve the quality of the paper. We hope very much that this manuscript, after the second round of revision, can now be published in the International Journal of Molecular Sciences.

Sincerely yours,

Filip Simunovic